# Molecular and Cellular Mechanisms of Vascular Development in Zebrafish

**DOI:** 10.3390/life11101088

**Published:** 2021-10-15

**Authors:** Jean Eberlein, Lukas Herdt, Julian Malchow, Annegret Rittershaus, Stefan Baumeister, Christian SM Helker

**Affiliations:** Cell Signaling and Dynamics, Faculty of Biology, Philipps-University Marburg, 35043 Marburg, Germany; jean.eberlein@biologie.uni-marburg.de (J.E.); lukas.herdt@biologie.uni-marburg.de (L.H.); julian.malchow@biologie.uni-marburg.de (J.M.); annegret.rittershaus@biologie.uni-marburg.de (A.R.); baumeist@staff.uni-marburg.de (S.B.)

**Keywords:** zebrafish, vasculogenesis, angiogenesis, blood vessels, endothelial cells, signaling pathways, Vegf, Apelin

## Abstract

The establishment of a functional cardiovascular system is crucial for the development of all vertebrates. Defects in the development of the cardiovascular system lead to cardiovascular diseases, which are among the top 10 causes of death worldwide. However, we are just beginning to understand which signaling pathways guide blood vessel growth in different tissues and organs. The advantages of the model organism zebrafish (*Danio rerio*) helped to identify novel cellular and molecular mechanisms of vascular growth. In this review we will discuss the current knowledge of vasculogenesis and angiogenesis in the zebrafish embryo. In particular, we describe the molecular mechanisms that contribute to the formation of blood vessels in different vascular beds within the embryo.

## 1. Introduction

The formation of a functional cardiovascular system is essential for the development of all vertebrates. During development, a network of blood vessels is formed to supply organs and tissues with oxygen, nutrients, signaling molecules and metabolites while simultaneously removing waste products contained in the blood. In addition to blood carrying vessels, the cardiovascular system is also comprised of a network of lymphatic vessels. These lymphatic vessels drain lymph from the capillary bed back to circulation and play a major role in fluid homeostasis and immunity [1].

The first artery and vein are formed de novo during embryonic development through differentiation of endothelial cell progenitors from the mesoderm in a process called vasculogenesis [2]. In contrast, during angiogenesis, new branches of blood vessels arise from pre-existing blood vessels [2]. The newly formed blood vessels then become lumenized and mature. The inner cell layer, facing the lumen, is formed by endothelial cells (ECs) that are enclosed by a basal lamina and mural cells (pericytes and smooth muscle cells) which regulate the vascular tone [3,4]. Unlike vasculogenesis, angiogenesis not only takes place during embryonic development, but is also reinitiated during wound repair/tissue regeneration [5] and several diseases, most notably cancer [4,6].

Since the introduction of the teleost fish *Danio rerio* (zebrafish) as a model system by Georg Streisinger in the 1970s [7], the zebrafish is a widely used model for analyzing developmental as well as vascular biology. As a vertebrate, 71% of human proteins have an orthologue in the zebrafish genome [8] and, most importantly, the signaling pathways that drive organ formation are conserved to mammals [9,10,11]. The rapid external development as well as large, synchronized clutches allow for the continuous monitoring of embryonic development at a large scale. Already at 24 h post fertilization (hpf) the major blood vessels are formed and heart contractility is initiated [12,13]. Whereas a defective cardiovascular system would be lethal in mammals, zebrafish larvae can survive up to five days without blood flow [14,15], making them an ideal model to study cardiovascular development. Already at 48 hpf the embryos develop the characteristic vertebrate body plan [12] and by pharmacological inhibition of pigmentation the embryos stay transparent [16], which enables the visualization of organ development even deep inside the tissue. Those properties, in addition to the diploid genome, make the zebrafish a valuable model for genetic [15,17,18] and pharmacological screens [11,19,20]. The first screens performed in zebrafish uncovered several genes that are important for the development of the cardiovascular system and other organs [15,17,18]. Moreover, the establishment of CRISPR/Cas9 as a standard method for gene editing in zebrafish [21] further simplified the analysis of gene function. In addition, the genetic accessibility facilitated the generation of transgenic lines that are shared within the community [22,23,24]. These transgenic lines enable precise observation of endothelial cell behavior, organelles, or even activities of enzymes [25] and second messengers like Ca^2+^ [26] in vivo.

## 2. Vasculogenesis in Zebrafish

During vasculogenesis, new blood vessels are formed de novo from mesoderm-derived endothelial progenitor cells, called angioblasts [2]. These angioblasts are initially specified within the lateral plate mesoderm (LPM) and start to migrate between the somites towards the embryonic midline [27]. Once the angioblasts reach the midline, they form the first circulatory loop consisting of the dorsal aorta (DA) and the cardinal vein [27,28,29,30]. So far, only a few pathways have been demonstrated to be required for vasculogenesis. In 1995, the *cloche* gene was identified as the first gene to be required for the specification of ECs [31]. Mutants for the *cloche* gene lack almost all endothelial as well as hematopoietic cells [31,32,33,34,35]. Hence, *cloche* is an upstream master-regulator, initiating signaling cascades driving the specification of endothelial and hematopoietic cell lineages [31]. Due to the location of the cloche gene on the telomere of chromosome 13, it took over two decades from the first identification of *cloche* to clone the gene *cloche*. Finally, the group of Didier Stainier, who first described the *cloche* mutant in 1995, identified *cloche* as the basic helix-loop-helix/Per-ARNT-SIM (bHLH-PAS) domain containing transcription factor *neuronal PAS domain protein 4 like* (*npas4l*) [36]. The expression of *npas4l* already starts during late gastrulation, where it acts upstream of the ETS1-related protein (Etsrp) and T-cell acute lymphocytic leukemia protein 1 (Tal1). [36].

Besides Npas4l, the ETS domain transcription factor Etsrp has been shown to be indispensable for vasculogenesis [37,38]. Expression analysis revealed that *etsrp* mRNA is already expressed in the LPM during the early somitogenesis in the zebrafish embryo [38,39]. Embryos deficient for *etsrp* exhibit a downregulation of endothelial-specific markers [38,40].

While EC specification is mediated by Npas4l and Etsrp function [31,35,36,37], the guidance of angioblasts towards the embryonic midline is mediated by Apelin receptor early endogenous ligand Apela (previously known as Elabela or Toddler), which signals through the G protein-coupled receptor Apelin receptor (Aplnr) [27]. Apela is secreted by the notochord and functions as a guidance cue for the angioblasts [27]. Depletion of both *aplnr* or both ligands *apela* and *apelin* (*apln*) resulted in an impaired angioblast migration [27].

## 3. Sprouting Angiogenesis in Zebrafish

### 3.1. Formation of the Intersegmental Vessels

At 48 hpf, blood is circulating through a series of aortic arches (Figure 1a,b), entering an anterior and posterior circulatory loop [13] (Figure 1a). The anterior vascular loop is connected to the brain vasculature (Figure 1a,c), whereas the posterior loop is connected to the DA [13] (Figure 1a,d). In the trunk, a series of intersegmental vessels (ISVs) is formed in-between the somites (Figure 1a,d). Arterial ISVs connect the DA to the dorsal longitudinal anastomotic vessel (DLAV) [13] (Figure 1a,d). Blood flows through the arterial ISVs into the DLAV and back through adjacent venous ISVs into the posterior cardinal vein (PCV) [13] (Figure 1a,d). Both the anterior and the posterior circulatory loop merge into the common cardinal vein (CCV), the largest vein in the embryo at this developmental timepoint [13,41] (Figure 1a,f). Blood then flows through the CCV over the yolk back to the heart, closing the circulatory loop [13,41].

During primary angiogenesis in the trunk of the zebrafish embryo, single endothelial cells within the DA sprout and migrate dorsally to form the ISVs (Figure 1a,d). This process requires a variety of signal cues. At around 19 hpf Vascular endothelial growth factor a (Vegfa) signaling through its receptor Kinase insert domain receptor like (Kdrl, previously known as Vegfr2 or Flk1) stimulates the sprouting of the ISVs in the zebrafish trunk. Binding of the ligand Vegfaa, which is expressed by the somites [42,43], modulates the phosphorylation of Kdrl. This in turn activates the Mitogen-activated protein kinase (Mapk) cascade and thus stimulates the phosphorylation of the Extracellular signal-regulated kinase (Erk) in a subset of ECs within the DA [44]. Zebrafish embryos which are deficient for either Vegfa or Kdrl exhibit an impaired ISV development [45,46,47].

Besides Kdrl, another Vegfr, termed the Fms Related Receptor Tyrosine Kinase 4 (Flt4, previously known as Vegfr3), is critical to promote angiogenic sprouting in zebrafish as well as in mammals [48,49]. However, in contrast to *kdrl* mutants, zebrafish mutants for *flt4* display defects in lymphatic and venous angiogenesis, while arterial development is mainly unaffected [50,51].

It has been recently reported that Tm4sf18, a member of the Transmembrane 4 L6 protein family, is activated by Vegfa signaling and is specifically expressed in sprouting ISVs [52]. A positive feedback loop between Tm4sf18 and Kdrl enhances Kdrl activity, which leads to robust sprouting of ISVs [52]. Consequently, the lack of Tm4sf18 function in zebrafish results in reduced Vegf signaling activity, vessel hypoplasia and truncated ISVs [52].

During sprouting angiogenesis, endothelial cells arrange as tip cells, leading the developing sprout and following stalk cells (Figure 2). Tip cells can be distinguished from stalk cells based on their morphology [53], expression of marker genes [54] and their metabolic state [55,56,57] (Figure 2). The determination of tip and stalk cells in ISVs is achieved by the Kdrl mediated downstream activation of Notch-Dll4 (Delta-like 4) signaling [49,58,59,60]. Vegf signaling in tip cells induces the expression of *dll4* [61]. Subsequently, Dll4 activates Notch signaling in adjacent stalk cells and represses the expression of both *dll4* and *kdrl* through lateral inhibition [49,58,59,60,61]. Thus, cells with reduced protein levels of Dll4 and Kdrl become stalk cells, whereas cells with active Vegf signaling are determined as tip cells [49,58,59,60]. In contrast, ECs deficient for Notch signaling exhibit a hyper = sprouting phenotype with more than one tip cell guiding the sprout [49,58,59,60]. However, recent studies showed that Notch signaling controls arterial angiogenesis by regulating the expression of C-X-C chemokine receptor type 4 (Cxcr4) and Vegfa [62,63]. In this model, Notch signaling is rather required in the tip cells and directs them into developing arteries [62].

During angiogenesis, both Kdrl and Flt4 exert their pro-angiogenic signaling by activating the MAPK cascade [44,64]. Inhibition of Erk activity prevents primary sprouting of ISVs and expression of *flt4*, as well as sprouting of the trunk lymphatics [44,64]. A recent study described a novel mechanism by which Vegfc/Flt4 signals via Erk induced G1 cell cycle arrest of lymphatic endothelial cells (LECs), thereby enhancing lymphatic sprouting efficiency [65]. In the ISVs Erk activity is higher in tip than in stalk cells, which is likely caused by higher Kdrl signaling in tip cells [25,64]. Interestingly, after division of the tip cell, this imbalance of Erk activity is maintained [25,66] due to the asymmetric cell division and thus partitioning of the *kdrl* mRNA [66].

While Kdrl and Flt4 are positive regulators of angiogenesis, Flt1 has been shown to restrict blood vessel growth [67,68]. In the zebrafish trunk, *flt1* is expressed in the developing vasculature, as well as in neurons [67,69,70]. Mutants for the soluble *flt1* (sFlt1) isoform exhibit a hyper-sprouting phenotype of venous ECs at the level of the neural tube [69,70]. Both, neuronal sFlt1 [69] or radial glia promoting sFlt1 expression in the ECs themselves [70] balance Vegfa signaling to modulate patterning of the trunk vasculature.

Further advancements in live imaging and transgenic tools uncovered Ca^2+^ oscillations during development of the ISVs [26,71] and capillaries of the brain [72]. During sprouting of the ISVs, Ca^2+^ oscillations are elevated in actively sprouting tip and stalk cells [26] and while Vegf signaling positively regulates calcium oscillations, Dll4/Notch signaling is required to suppress calcium oscillations in ECs adjacent to the stalk cell [26]. The transmembrane protein 33 (Tmem33), a three-pass transmembrane domain protein located on the endoplasmic reticulum, has been demonstrated to be required for Ca^2+^ oscillations in response to Vegf [71]. Zebrafish *tmem33* mutants exhibit a reduced number of filopodia and impaired sprouting of ISVs [71]. Calcium signaling activity was also found to be important for vascular pathfinding in the developing zebrafish brain [72]. Here, high and low frequency calcium oscillations are modulated by the mechanosensitive channel Piezo1 and regulate extension or retraction of vascular branches [72].

Another pathway that has emerged as an important regulator of angiogenesis is the Apelin signaling pathway. In contrast to *vegfaa*, which is expressed in the somites [42,43] and neuronal cells [69], *apln* expression can be observed in the developing ISVs and is enriched in tip cells [54,56]. In zebrafish, the Apelin pathway consists of two peptide ligands Apela and Apelin that signal through the G protein-coupled receptors Aplnra and Aplnrb. As mentioned in Section 2, Apela mainly regulates angioblast migration during vasculogenesis [27]. In contrast, Apelin mediated activation of Apelin receptors is required for ISV sprouting [56]. Interestingly, the expression of *apln* is downregulated in mature blood vessels, but is reactivated in sprouting ECs during regeneration [73] and tumor angiogenesis [74,75,76]. Furthermore, *apln* expression is regulated by Notch signaling and Apelin deficiency prevents, similar to the knockdown of *flt4*, the hyper-sprouting phenotype induced by knockdown of *dll4* [50,56]. To facilitate sprouting, Apelin signaling positively regulates EC metabolism by modulating the expression of the pro-metabolic proteins PFKFB3 and C-MYC in human umbilical vein endothelial cells (HUVECs) [56].

Whereas Vegf and Apelin signaling are important pathways involved in ISV sprouting, repulsive signals by Semaphorin-Plexin signaling restrict the migration path of growing ISVs to the somite boundary [77]. While the *plexinD1* receptor is specifically expressed by endothelial cells, its ligand *semaphorin3ab (sema3ab)* is expressed by the surrounding somites and thereby prevents ECs invasion into the somite [77]. Consequently, *plexinD1* mutant embryos exhibit ectopic ISV sprouts which exceed the somite boundary [77,78].

In the zebrafish trunk, neighboring ISVs fuse in a process called anastomosis to form the DLAV at around 30 hpf. Initially, all ISV sprouts are arterially derived. However, venous sprouts emerging from the PCV migrate dorsally to contact the arterial ISVs. This process is driven by *vegfc* expression by ECs in the DA and *flt4* expression by ECs in the PCV [79]. Venous sprouts then fuse to the arterial ISVs and migrate against the direction of blood flow, displacing arterial ECs and transforming the arterial ISV into a venous ISV [80] (Figure 3a). Activation of Notch signaling in some ISVs protects those ISVs from being transformed into venous ISVs [80,81]. Some 65% of the venous sprouts express the transcription factor Prox1 [82], and 50% of these will become LECs forming the parachordal lymphangioblasts (PLs) or parachordal cells (PACs) [83] (Figure 3a). These LECs will continue to migrate along the arteries to form the main lymphatic vessels in the trunk of the zebrafish [83] (Figure 3b).

Endothelial cell-to-extracellular matrix (ECM) interactions play an important role during sprouting as well as maintenance of blood vessels. During sprouting, ECs adhere to the ECM to facilitate their movement [84]. In addition, perivascular fibroblasts express ECM components, such as the collagens *col1a2* and *col5a1* to stabilize blood vessels [85].

### 3.2. Formation of the Caudal Vein

The caudal vein (CV) is located in the caudal part of the PCV, starting with the end of the yolk extension [13]. At around 27 hpf, ECs of the CV sprout ventral and start to form the caudal vein plexus (CVP) [86]. This ventral sprouting of venous ECs is highly dependent on bone morphogenetic protein (BMP) signaling [86,87], while Vegfa signaling has no effect on ventral sprouting of venous ECs [87]. Vice versa, arterial angiogenic sprouts do not respond to BMP signaling, but are highly dependent on Vegfa signaling [87]. Downstream of BMP signaling, Cdc42 activates Formin-like 3 (Fmnl3), an actin-regulating protein, leading to filopodia extension to facilitate angiogenic sprouting of the CVP [86]. Inhibition of filopodia formation by either Latrunculin A treatment or injection of a *fmnl3* morpholino (MO) leads to defects in the development of the CVP [86,88]. During ventral sprouting venous ECs exhibit high β-Catenin-dependent transcriptional activity [89]. Disturbance of β-Catenin-dependent transcription, either by overexpression of *axin* or a dominant-negative T-cell factor (Tcf), results in impaired CV formation [89]. By 36 hpf ventral sprouting of ECs led to the formation of a vascular plexus, named CVP. ECs in the ventral part of the CVP maintain high β-Catenin activity and form the lumenized definitive CV by 48 hpf [89].

The formation of the CV-primordium, as well as sprouting from the CV, are both dependent on hemodynamic forces by blood flow [90,91,92]. Blocking cardiac contraction and thereby blood flow in zebrafish embryos, either by treatment with ion channel blockers (nifedipine, BDM and tricaine) or by injection of troponin T type 2a (*tnnt2a*) MO, results in an impaired CVP formation [90,91,92].

### 3.3. Formation of the Brain Vasculature

Among all organs, the brain has the highest oxygen and nutrient consumption. Hence, a complex network of blood vessels in the brain is necessary to ensure an appropriate supply of oxygen and nutrients [93,94]. The vascularization of the brain is initiated at 32 hpf by angiogenic sprouting from the primordial hindbrain channels (PHBC) forming the central arteries (CtA) (Figure 1c) [95,96,97]. Subsequently, CtA sprouts anastomose with the basilar artery (BA) and lumenize between 36 and 48 hpf (Figure 1c) [95,96,97]. In contrast, the mesencephalic CtAs (MCtA) in the midbrain sprout from the perineural vascular plexus (PVP) into the brain parenchyma at 36 hpf (Figure 1c) [13,98]. Once sprouting and perfusion of the brain is completed, ECs start to establish the blood-brain-barrier (BBB) [99,100,101]. The BBB controls the selective transport of molecules into the brain parenchyma, while it also protects the brain by limiting the entry of pathogens and harmful molecules (reviewed in [102,103]).

Similar to its role during the formation of the trunk vasculature, Vegf signaling plays a crucial role for the development of the cerebral vasculature. Zebrafish embryos treated with SU5416, an inhibitor of the tyrosine kinase activity of all Vegf receptors, exhibit defects in the formation of the BA, CtAs, PHBC and mid-cerebral vein (MCeV) [47,97]. In particular, Vegfa signaling is essential for sprouting of the CtAs in the hindbrain as demonstrated by the absence of CtAs in mutants for *kdrl* and *vegfaa* [96]. Furthermore, the combination of *vegfab*, *vegfc* and *vegfd* is required for the development of fenestrated vessels in the myelencephalic choroid plexus (mCP) [104]. Triple mutants for *vegfab*, *vegfc* and *vegfd* exhibit impaired mCP vascularization, while the formation of non-fenestrated brain vessels was unaffected [104].

CXC-receptors are G protein-coupled receptors that are activated by small CXC-motif containing chemokines (as reviewed in [105,106]). CXC chemokines function as guidance cues for migrating cells [107,108,109]. During brain angiogenesis of the zebrafish embryo, Cxcr4-Cxcl12 signaling plays a crucial role for the vascularization of the hindbrain. In embryos deficient for the receptor *cxcr4a* or ligand *cxcl12b*, sprouting of the CtAs is unaffected [96]. However, both mutants exhibit defects in pathfinding of the CtAs towards the BA, resulting in an increased number of unperfused CtA [96,97]. In contrast, sprouting and pathfinding of intersegmental vessels in the zebrafish trunk is unaffected in both *cxcl12b* and *cxcr4a* mutants, highlighting the distinct role of Cxcr4 signaling during hindbrain vascularization [96]. Subsequently, perfusion of the CtAs leads to a downregulation of *cxcr4a* expression, while unperfused CtAs maintain high *cxcr4a* expression levels [96], indicating that *cxcr4a* mRNA expression is negatively regulated by blood flow.

Canoncial Wnt/β-Catenin signaling plays a pivotal role both during brain angiogenesis and BBB formation [110,111,112]. In sprouting ECs of the brain, active Wnt/β-Catenin signaling counteracts Sphingosine-1-phosphate receptor 1-(S1pr1) induced BBB formation but decreases after lumen formation [112]. This decrease in Wnt/β-Catenin signaling then leads to the activation of S1pr1-signaling regulating the localization of the cell-cell junction molecules Ve-cadherin and Esama and therefore BBB formation [112].

The Adhesion G protein-coupled receptor A2 (Adgra2, previously known as Gpr124), is a membrane-bound G protein-coupled receptor that plays a key role in the development of the brain vasculature in zebrafish [113]. These *adgra2* mutants exhibit vascular defects in the brain, especially in the formation of the CtAs [113]. Together with the GPI-anchored protein Reversion-inducing cysteine-rich protein with Kazal motifs (Reck), Adgra2 forms a receptor complex that promotes Wnt7a/Wnt7b-dependent canonical Wnt/β-Catenin signaling during sprouting of the CtAs in the brain [113,114,115]. Here, Adgra2 binds to the extracellular domain of Reck to enable the formation of a complex consisting of Adgra2, Reck, Frizzled (Fz) and Lrp5/6 [114]. This complex is required to deliver Reck-bound Wnt7 to the Frizzled receptors [114]. In addition, Adgra2 binds to the intracellular domain of the scaffolding protein Dishevelled (Dvl) [114]. Dvl functions as a bridge for Fz and Adgra2 to trigger Wnt/β-Catenin signaling through the Fz receptor and Lrp5/6 co-receptors [114]. Moreover, transplantation experiments of wildtype endothelial cells into *adgra2* deficient embryos revealed that Adgra2/Reck is specifically required in CtA tip cells but not ISV tip cells [113].

## 4. Lumen Formation

### 4.1. Cord Hollowing and Cell Hollowing

A crucial step in blood vessel morphogenesis is the transition from an early sprout to a tubular structure that enables fluid transport. The lumen of these tubes is formed either extracellularly (cord hollowing or budding), opening up a lumen in-between the ECs [29,116,117], or intracellularly by “cell hollowing”, forming a lumen within the cell [118]. During cord hollowing, ECs align to each other as a cord, which can be observed by the presence of cell-cell contacts [29,116,117]. Subsequently, ECs open the lumen in between them [29,116,117]. The concept of cell hollowing [118] has been proposed in particular for lumen formation during anastomosis. Here, the pressure from blood flow leads to tunnel- like membrane invaginations at the distal growing end of the sprouting vessels leading to the formation of a lumen within a single cell [118]. Recently, it was shown that this process is accompanied by transient recruitment and contraction of actomyosin resulting in inverse membrane blebbing, which appears to be dependent on differences between intra and extracellular pressure [119]. Once blood flow is established within the new vessel, ECs align in the direction of blood flow and reorient their Golgi apparatus against the direction of blood flow [120]. This process has been shown to be dependent on Aplnr signaling via ß-Arrestin [120].

### 4.2. Alternative Ways of Lumen Formation

A novel mechanism of extracellular lumen formation, named lumen ensheathment, was shown for the formation of the CCV [41]. Here individual ECs align in a sheath-like manner around an initially virtual tube lumen [41]. Another way of lumen formation has recently been described for the CV-primordium [90]. Together with the CCV, the CV-primordium is one of the largest vessels observed during embryonic development, being 5 times larger than the DA [47,90]. At 18 hpf endothelial struts coalesce in the future lumen of the CV [90]. Consequently, single ECs start to form the vessel wall around the network of endothelial struts [90]. At around 26–28 hpf, endothelial struts prune and integrate into the vessel wall [90]. Laser ablation of endothelial cell struts results in a collapse of the CV upon circulation, indicating that these elements provide structural support to the CV [90]. Inhibition of BMP signaling causes a failure of ECs to coalesce into struts, leading to defective CV development [90]. Conversely, global overexpression of *bmp2b* leads to pruning defects of endothelial struts, which are still remaining at 48 hpf, resulting in an only partially formed CV wall [90]. Along with the mechanism of lumen ensheathment [41], endothelial struts [90] provide another way for ECs to form large-caliber blood vessels.

## 5. Vascular Remodeling

During organ growth, the vascular system is permanently adapting to the changing requirements of the growing organism by remodeling of the vascular network [121]. Through remodeling of the vascular network, shortcuts and loops are removed to ensure an optimal and unidirectional blood flow that efficiently supplies all organs with [98,122]. On the cellular level, pruning is reminiscent of anastomosis in reverse, i.e., the pruning segment reduces its lumen diameter until the lumen collapses and cell-cell contacts are removed until ECs separate and migrate back into the adjacent vessel branches [98,123]. The selection of vessels to be pruned is triggered by low or fluctuating blood flow and in this way supports stabilized segments with constant blood flow [98,123,124,125,126]. On the molecular level it has been shown that the pruning process is accompanied by an activation of Rac1 triggered by low blood flow, which contributes to the increased migratory capacity of ECs [98]. Recently, *klf6a* and *tagln2* were identified to regulate cell-cell contacts and cytoskeleton rearrangement to support pruning of the caudal vein [127].

## 6. Development of Hematopoietic Stem Cells

### 6.1. Emergence of Hematopoietic Stem Cells

Hematopoietic stem cells (HSCs) generate all blood lineages during adult life [128,129]. The hemogenic endothelium (HE), a specialized subpopulation of endothelial cells (EC), located in the ventral floor of the DA (Figure 4a), generates HSCs during development in a process known as endothelial-to-hematopoietic transition (EHT) [130,131,132,133]. EHT is controlled by a variety of signaling pathways such as Notch-[134,135], TGFβ [136,137], BMP [138], Wnt9a/β-Catenin [139], Hif [140,141], YAP [142] and NOS-signaling [143]. Following their specification within the HE, nascent HSCs enter circulation and colonize the CVP [144,145] (Figure 4b).

### 6.2. Homing of Hematopoietic Stem Cells

The vascular plexus of the CVP is also known as caudal hematopoietic tissue (CHT), which is the zebrafish equivalent to the mammalian fetal liver [145] and serves as a vascular niche for HSCs [144,145,146,147,148]. Upon emerging from the ventral floor of the DA (Figure 4a), HSCs enter the circulation [130,131,144]. From 48 hpf onwards circulating HSCs colonize the CHT [146], where they interact with Vcam1-expressing macrophages [149]. This interaction between HSCs and Vcam1-expressing macrophages requires Integrin alpha 4 and is crucial for homing of HSCs to small venous capillaries [149]. Next, HSCs extravasate to the abluminal side of the vascular wall and trigger the surrounding ECs to form a pocket like structure in a process called “endothelial cuddling” [146]. Within the pocket like structure HSCs dynamically interact with mesenchymal stromal cells (MSC) [146]. The complex environment within the niche, created by ECs, MSCs and also immune cells, like macrophages and neutrophils, tightly regulates HSC proliferation, differentiation and egression of the HSCs from the CHT from 72 hpf onward [145,148,150]. After leaving the vascular niche HSCs eventually populate the adult lymphopoietic- and hematopoietic organs, the thymus and the kidney in zebrafish [144,145].

## 7. Perspective

Over the past four decades, experiments conducted in zebrafish have contributed to a deeper understanding of the morphogenetic processes that shape the vertebrate body. The initially simple blueprint of the zebrafish vasculature allows for fast gain and loss of function analyses. The optical clarity of the zebrafish embryo in combination with transgenic techniques enables the analysis of the behavior of cells and signaling processes live at a single cell resolution using confocal microscopy. The development of new imaging technologies, such as light sheet microscopy, new fluorophores and biosensors, will allow even more detailed images to be acquired over longer periods of time without bleaching or photo toxicity. Technologies such as small molecule screens [151,152,153] and protein-protein interaction assays (e.g., BioID) [154,155] are extending the toolbox for using the zebrafish model. The emergence of single cell sequencing technologies within the last years enabled the dissection of the developmental trajectories of cells and their heterogeneity within one cell lineage [156].

## Figures and Tables

**Figure 1 life-11-01088-f001:**
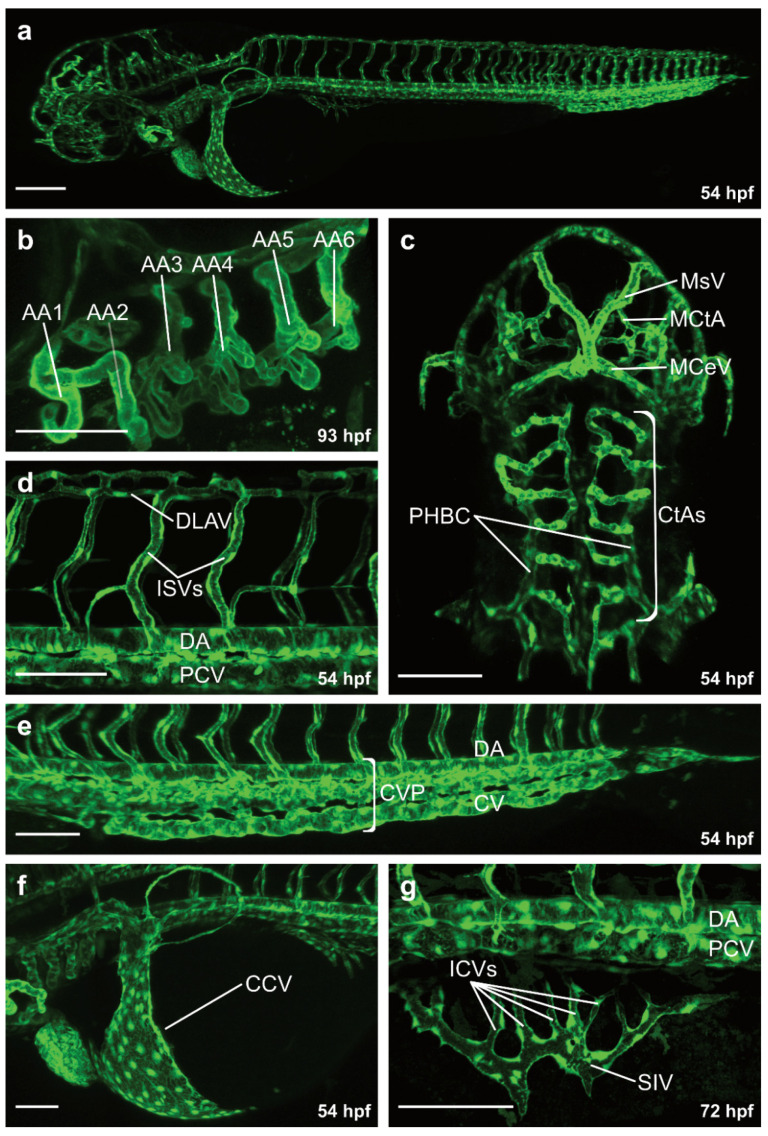
Overview of the vasculature of transgenic zebrafish larvae. Confocal projection images of the vasculature in lateral (**a**,**b**,**d**–**g**) and dorsal view (**c**). (**a**) Overview of the vasculature at 54 hpf; (**b**) Magnification of the aortic arches at 93 hpf; (**c**) Magnification of the blood vessels in the brain at 54 hpf; (**d**) Magnification of the trunk vasculature at 54 hpf; (**e**) Magnification of the posterior trunk vasculature including dorsal aorta, caudal vein plexus and caudal vein at 54 hpf; (**f**) Magnification of the common cardinal vein at 54 hpf; (**g**) Magnification of the subintestinal vein plexus at 72 hpf. AA, aortic arches; CtA, central artery; MsV, mesencephalic vein; MCeV, mid-cerebral vein; MCtA, mesencephalic central artery; DLAV, dorsal longitudinal anastomotic vessel; ISV, intersegmental vein; DA, dorsal aorta; PCV, posterior cardinal vein; CVP, caudal vein plexus; CCV, common cardinal vein; SIV, subintestinal vein; ICV, interconnecting vessel; Scale bars: overview image 200 µm; all magnifications 100 µm.

**Figure 2 life-11-01088-f002:**
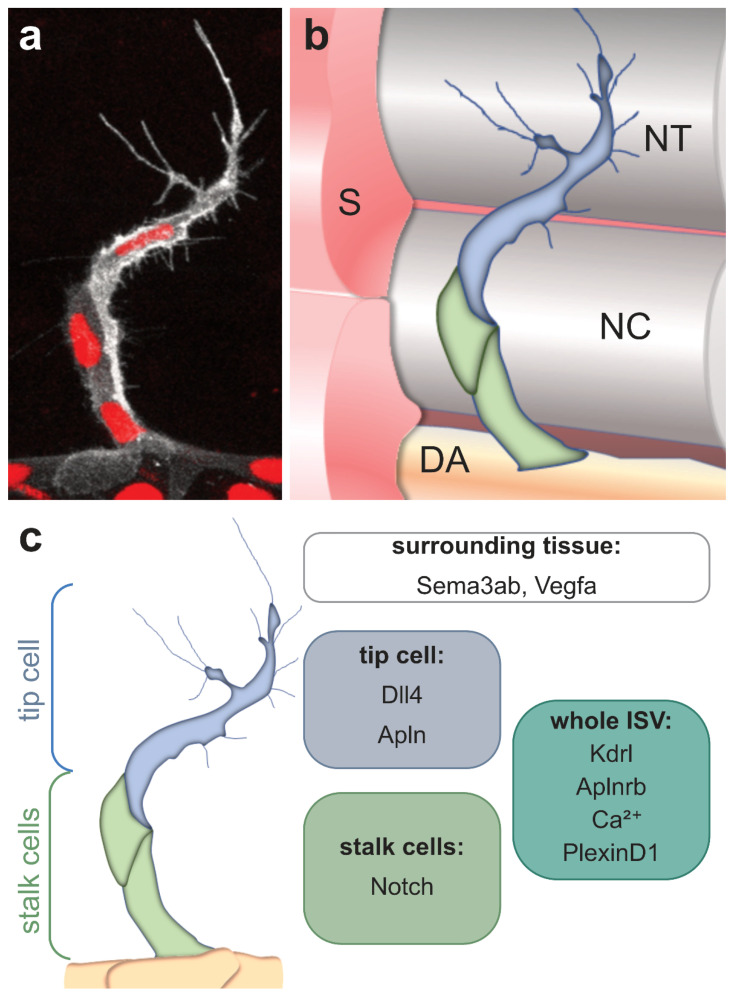
Schematic overview of a sprouting intersegmental vessel. (**a**) Magnification of a single ISV in a double transgenic zebrafish embryo. The cell membrane is visualized in white (*Tg(kdrl:GFP-CAAX)*) and the nuclei are labeled in red (*Tg(kdrl:NLS-mCherry)*). Note the long cell protrusions/filopodia of the tip cell, while the following stalk cells only exhibit short filopodia; (**b**) Schematic overview of a sprouting ISV and the surrounding tissues. The tip cell is labeled in blue and the stalk cells in green. ECs of the ISV sprout from the dorsal aorta (DA) to migrate at the somite boundary around the notochord (NC), eventually leading to the formation of the dorsal longitudinal anastomotic vessel (DLAV) at the level of the neural tube (NT); (c) The tip cell and stalk cells can be distinguished morphologically and by the expression of marker genes. While the whole ISV is expressing Kdrl, Aplnrb, PlexinD1 and exhibit Ca^2+^ oscillations, only the tip cell is expressing high levels of the Notch ligand *dll4* and the ligand *apln*. In contrast, the Notch receptor is highly expressed by the stalk cells. In addition, the surrounding tissue is also providing attractive signals such as *vegfa* and repulsive signals such as *semaphorin* ligands. S, somite; NT, neural tube; NC, notochord; DA, dorsal aorta.

**Figure 3 life-11-01088-f003:**
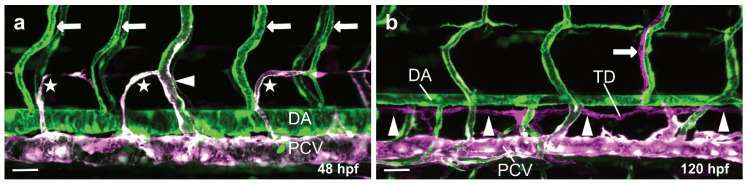
Overview of arterial, venous and lymphatic vessels in the trunk of transgenic zebrafish larvae. Confocal projection images of the trunk vasculature of a double transgenic zebrafish line (*Tg(kdrl:EGFP); Tg(-5.2lyve1b:DsRed)*) at 48 hpf (**a**) and 120 hpf (**b**) in lateral view. Arterial endothelial cells (ECs) are labeled in green (GFP), venous ECs are labeled in green and magenta (GFP and DsRed double positive) and lymphatic ECs are labeled in magenta (DsRed). (**a**) Magnification of a zebrafish trunk showing the DA and arterial (arrows) and venous ISVs (arrowhead) as well as lymphatic sprouts (stars); (**b**) Magnification of a zebrafish trunk showing the main lymphatic vessels: the thoracic duct (TD) (arrowheads) and one intersegmental lymphatic vessel (ISLV) (arrow); DA, dorsal aorta; PCV, posterior cardinal vein; ISV, intersegmental vessel; TD, thoracic duct; Scale bars: 30 µm.

**Figure 4 life-11-01088-f004:**
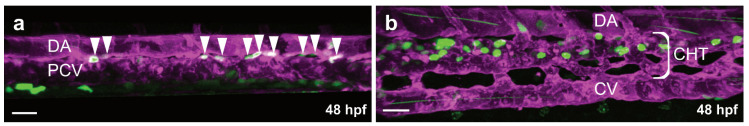
Hematopoietic stem cell development in transgenic zebrafish larvae. Confocal projection images of the trunk (**a**) and the tail (**b**) of a 48 hpf zebrafish embryo in lateral view. HSCs are visualized in green (*Tg(itga2b:GFP)*) and the vasculature in magenta (*Tg(kdrl:HsHRAS-mCherry)*). (**a**) HSCs (green) emerging from the DA (arrowheads); (**b**) HSCs (green) residing in the CHT. HSCs, Hematopoietic stem cells; DA, dorsal aorta; PCV, posterior cardinal vein; CHT, caudal hematopoietic tissue; scale bars: 30 µm.

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
