# Peer review of "Molecular and Cellular Mechanisms of Vascular Development in Zebrafish"

_life, 2021, doi:10.3390/life11101088_

Round 1

Reviewer 1 Report

In this review, the current knowledge of vasculogenesis and angiogenesis on zebrafish development is discussed. Authors particularly describe the molecular mechanisms that contribute to the formation of blood vessels in different vascular beds. It is recognized the interest and relevance of the subject, as zebrafish early life can act as a proxy of vertebrate’s development (within the same time-window). Manuscript is fine written and reading was relatively easy to follow. Although not a systematic review (that would be preferable), the manuscript composes a fine structured review of the topic, supported by relevant references. However, few statements requiring literature support were highlight. Moreover, it is recommended to add as last section, one brief paragraph dedicated to "Future studies/directions" highlighting the important gaps to fulfill and discussing what/how would be interesting to assess these. Figures and Tables are pertinent and detailed. Minor questions/concerns were highlighted through the manuscript document enclosed.

Author Response

Dear MDPI Life Editorial Team,

Please find attached our revised manuscript entitled “Molecular and Cellular Mechanisms of Vascular Development in Zebrafish”, for consideration as a review article in the Advances in Zebrafish Genetics and Applications to Study the Cardiovascular System Special Issue of MDPI Life. We thank you for the initial positive feedback and for giving us the opportunity to resubmit. We deeply appreciate your comments and those of the reviewers and as a consequence, we have made significant changes to the manuscript in order to respond to these constructive critiques.

Please find below our detailed point-by-point response to the reviewers’ comments.

Thank you for considering our manuscript.

Kind regards,

Christian Helker

Reviewer 1:

In this review, the current knowledge of vasculogenesis and angiogenesis on zebrafish development is discussed. Authors particularly describe the molecular mechanisms that contribute to the formation of blood vessels in different vascular beds. It is recognized the interest and relevance of the subject, as zebrafish early life can act as a proxy of vertebrate’s development (within the same time-window). Manuscript is fine written and reading was relatively easy to follow. Although not a systematic review (that would be preferable), the manuscript composes a fine structured review of the topic, supported by relevant references. However, few statements requiring literature support were highlight. Moreover, it is recommended to add as last section, one brief paragraph dedicated to "Future studies/directions" highlighting the important gaps to fulfill and discussing what/how would be interesting to assess these. Figures and Tables are pertinent and detailed. Minor questions/concerns were highlighted through the manuscript document enclosed.

We thank the reviewers for these valid points. We changed all mentioned points in the manuscript and added a perspective in the end of the manuscript.

Reviewer 2 Report

The authors have performed a great service in summarizing the current state of research into zebrafish vascular development across the relevant levels of organization, from phenomenological, to cellular and molecular. The manuscript is generally well-written, and the organization and logic are clear. The expertise of the author in this field is obvious in the writing. In my opinion, the manuscript could be slightly improved with additional information about the role of the extracellular matrix for guidance in pathfinding, vascular remodeling during regeneration (briefly).

-Line 102, Spouting should be Sprouting

Author Response

Dear MDPI Life Editorial Team,

Please find attached our revised manuscript entitled “Molecular and Cellular Mechanisms of Vascular Development in Zebrafish”, for consideration as a review article in the Advances in Zebrafish Genetics and Applications to Study the Cardiovascular System Special Issue of MDPI Life. We thank you for the initial positive feedback and for giving us the opportunity to resubmit. We deeply appreciate your comments and those of the reviewers and as a consequence, we have made significant changes to the manuscript in order to respond to these constructive critiques.

Please find below our detailed point-by-point response to the reviewers’ comments.

Thank you for considering our manuscript.

Kind regards,

Christian Helker

Reviewer 2:

The authors have performed a great service in summarizing the current state of research into zebrafish vascular development across the relevant levels of organization, from phenomenological, to cellular and molecular. The manuscript is generally well-written, and the organization and logic are clear. The expertise of the author in this field is obvious in the writing. In my opinion, the manuscript could be slightly improved with additional information about the role of the extracellular matrix for guidance in pathfinding, vascular remodeling during regeneration (briefly).

-Line 102, Spouting should be Sprouting

We thank the reviewers for these valid points. We added a new paragraph on the role of the ECM during vascular development.